# *Acinetobacter* Non-*baumannii* Species: Occurrence in Infections in Hospitalized Patients, Identification, and Antibiotic Resistance

**DOI:** 10.3390/antibiotics12081301

**Published:** 2023-08-09

**Authors:** Eugene Sheck, Andrey Romanov, Valeria Shapovalova, Elvira Shaidullina, Alexey Martinovich, Natali Ivanchik, Anna Mikotina, Elena Skleenova, Vladimir Oloviannikov, Ilya Azizov, Vera Vityazeva, Alyona Lavrinenko, Roman Kozlov, Mikhail Edelstein

**Affiliations:** 1Institute of Antimicrobial Chemotherapy, Smolensk State Medical University, 214019 Smolensk, Russia; evgeniy.sheck@antibiotic.ru (E.S.); ilya.azizov@antibiotic.ru (I.A.);; 2Republican Children’s Hospital, 185000 Petrozavodsk, Republic of Karelia, Russia; 3Shared Resource Laboratory, Karaganda Medical University, 100008 Karaganda, Kazakhstan

**Keywords:** *Acinetobacter* non-*baumannii*, MALDI-TOF MS, *rpoB* gene sequencing, antibiotic resistance, carbapenemases

## Abstract

Background: *Acinetobacter* species other than *A. baumannii* are becoming increasingly more important as opportunistic pathogens for humans. The primary aim of this study was to assess the prevalence, species distribution, antimicrobial resistance patterns, and carbapenemase gene content of clinical *Acinetobacter* non-*baumannii* (*Anb*) isolates that were collected as part of a sentinel surveillance program of bacterial infections in hospitalized patients. The secondary aim was to evaluate the performance of MALDI-TOF MS systems for the species-level identification of *Anb* isolates. Methods: Clinical bacterial isolates were collected from multiple sites across Russia and Kazakhstan in 2016–2022. Species identification was performed by means of MALDI-TOF MS, with the Autobio and Bruker systems used in parallel. The PCR detection of the species-specific *bla*_OXA-51-like_ gene was used as a means of differentiating *A. baumannii* from *Anb* species, and the partial sequencing of the *rpoB* gene was used as a reference method for *Anb* species identification. The susceptibility of isolates to antibiotics (amikacin, cefepime, ciprofloxacin, colistin, gentamicin, imipenem, meropenem, sulbactam, tigecycline, tobramycin, and trimethoprim–sulfamethoxazole) was determined using the broth microdilution method. The presence of the most common in *Acinetobacter*-acquired carbapenemase genes (*bla*_OXA-23-like_, *bla*_OXA-24/40-like_, *bla*_OXA-58-like_, *bla*_NDM_, *bla*_IMP_, and *bla*_VIM_) was assessed using real-time PCR. Results: In total, 234 isolates were identified as belonging to 14 *Anb* species. These comprised 6.2% of *Acinetobacter* spp. and 0.7% of all bacterial isolates from the observations. Among the *Anb* species, the most abundant were *A. pittii* (42.7%), *A. nosocomialis* (13.7%), the *A. calcoaceticus/oleivorans* group (9.0%), *A. bereziniae* (7.7%), and *A. geminorum* (6.0%). Notably, two environmental species, *A. oleivorans* and *A. courvalinii*, were found for the first time in the clinical samples of patients with urinary tract infections. The prevalence of resistance to different antibiotics in *Anb* species varied from <4% (meropenem and colistin) to 11.2% (gentamicin). Most isolates were susceptible to all antibiotics; however, sporadic isolates of *A. bereziniae*, *A. johnsonii*, *A. nosocomialis*, *A. oleivorans*, *A. pittii*, and *A. ursingii* were resistant to carbapenems. *A. bereziniae* was more frequently resistant to sulbactam, aminoglycosides, trimethoprim–sulfamethoxazole, and tigecycline than the other species. Four (1.7%) isolates of *A. bereziniae*, *A. johnsonii*, *A. pittii* were found to carry carbapenemase genes (*bla*_OXA-58-like_ and *bla*_NDM_, either alone or in combination). The overall accuracy rates of the species-level identification of *Anb* isolates with the Autobio and Bruker systems were 80.8% and 88.5%, with misidentifications occurring in 5 and 3 species, respectively. Conclusions: This study provides important new insights into the methods of identification, occurrence, species distribution, and antibiotic resistance traits of clinical *Anb* isolates.

## 1. Introduction

The genus *Acinetobacter* belongs to the family *Moraxellaceae*, class γ-proteobacteria, and comprises coccobacillary-shaped Gram-negative, aerobic, non-lactose fermenting, saprophytic bacteria. This genus has undergone substantial taxonomic modification and currently comprises 82 species with valid published names and 24 species with non-verified published or provisionally assigned names (https://lpsn.dsmz.de/genus/acinetobacter, accessed on 16 July 2023) [1]. Most *Acinetobacter* species are ubiquitous in the environment (soil, water, plants, and animals), and some have evolved as important opportunistic pathogens for humans and animals and have adapted to cause various infections, especially in compromised hosts [2]. *A. baumannii* is a primary human pathogen and is one of the main causes of nosocomial infections with the highest mortality rates [3,4]. Its intrinsic resistance to many antibiotics and its remarkable ability to acquire resistance to all available therapeutic agents, including carbapenems, have secured it a place in the group of ESKAPE pathogens (*Enterococcus faecium*, *Staphylococcus aureus*, *Klebsiella pneumoniae*, *Acinetobacter baumannii*, *Pseudomonas aeruginosa*, and *Enterobacter* spp.) [5] and a top place in the WHO’s global priority list of antibiotic-resistant bacteria [6]. *A. baumannii* and seven other closely related species (*A. calcoaceticus*, *A. geminorum* (the most recently described species), *A. lactucae* (formerly also known as *A. dijkshoorniae*), *A. nosocomialis*, *A. oleivorans* (effectively but not validly published named species), *A. pittii*, and *A. seifertii*) together form the *Acinetobacter calcoaceticus-baumannii* (*Acb*) complex [7,8,9,10]. Species within the *Acb* complex are almost indistinguishable phenotypically but differ significantly in terms of their ecology, pathogenicity, epidemiology, and susceptibility to antibiotics [11]. As an example, while all *Acb* species are capable of causing infections in humans, *A. calcoaceticus* and *A. oleivorans* are primarily environmental species and are less frequently isolated from human clinical specimens [9,12]. In contrast, *A. pittii* and *A. nosocomialis* are more often associated with hospital-acquired infections [11]. As well as *Acb*, 20 more species of the genus *Acinetobacter* were included in the recently updated list of bacteria cultured from humans [13]; among them, *A. bereziniae*, *A. johnsonii*, *A. junii*, *A. lwoffii*, *A. soli*, and *A. ursingii* have commonly been reported in human infections. Although significantly less prevalent than *A. baumannii* and typically less resistant to antibiotics, *Acinetobacter* non-*baumannii* (*Anb*) species are becoming increasingly more important as nosocomial pathogens [14,15,16,17,18,19,20].

The accurate identification of *Anb* species is therefore crucial but remains challenging, especially within the *Acb* complex [21]. In the last two decades, molecular methods have been introduced to enable the more efficient differentiation of *Acinetobacter* species as compared to conventional phenotypic identification by manual biochemical tests or automated systems. The sequencing of certain protein-encoding genes (*rpoB* for RNA polymerase subunit B, which is used the most often, *gyrB* for DNA gyrase subunit B, or *recA* for DNA repair recombinase) and/or multilocus sequence analysis (MLSA) have proven the most accurate methods and thus constitute the current reference standard for the identification of *Acinetobacter* species [8,21,22,23]. The PCR detection of *A. baumannii* species-specific genes for class D β-lactamase (*bla*_OXA-51-like)_ has also been used as a quick and effective means of differentiating *A. baumannii* from *Anb* species [24]. The use of matrix-assisted laser desorption/ionization time-of-flight mass spectrometry (MALDI-TOF MS) for the identification of bacterial species has been a major breakthrough in clinical microbiology. It has proven to be a rapid, cost-effective, and accurate method of differentiating even closely related bacterial species that are otherwise indistinguishable by conventional phenotypic methods [25]. Several studies have assessed the performance of MALDI-TOF MS systems for the identification of *Acinetobacter* species, and, notably, the results from these studies have demonstrated the crucial importance of the size and quality of reference spectra databases (libraries) for accurate species-level identification [21,26,27].

The epidemiology and mechanisms of antibiotic resistance have been most extensively studied for *A. baumannii*. In particular, numerous publications have addressed the problem of the global spread of carbapenem-resistant strains and acquired carbapenemase-producing strains of this species, which pose the most significant health threat [3,4,28]. Other *Acinetobacter* species have been less well studied, although there have been many reports from many parts of the world of isolates of different *Anb* species producing different carbapenemases [29,30,31,32,33,34,35,36,37,38,39,40,41,42]. A recent analysis of publicly available genome sequences showed that *Anb* species represent a reservoir of many transferrable resistance genes to different classes of antibiotics, other than carbapenems [43].

Therefore, this study specifically focused on clinical *Anb* isolates that were collected as part of a large sentinel surveillance program in 2016–2022. We aimed to assess: (i) the prevalence of *Anb* in infections in hospitalized patients; (ii) the performance of MALDI-TOF MS systems for the species-level identification of *Anb* in comparison with the reference *rpoB* sequencing method; and (iii) the prevalence of antibiotic resistance and acquired carbapenemases in *Anb* species.

## 2. Results

### 2.1. Species Distribution of Acinetobacter Isolates

A total of 3754 (10.9%) of the bacterial isolates collected in 2016–2022 within the frame of AMR sentinel surveillance program in Russia and Kazakhstan were identified as members of the *Acinetobacter* genus. *A. baumannii* was the most prevalent species, comprising 3520 (10.2%) isolates, while the other *Acinetobacter* species jointly comprised 234 (0.7%) bacterial isolates. The *Anb* isolates were collected from 41 hospitals in 27 cities across Russia and Kazakhstan. Using the phylogenetic clustering of *rpoB* gene sequence data, they were assigned to 14 different species (Figure 1). The clinical isolates clustered well with the corresponding reference strains and the clusters of different species were clearly separated on the tree. The only exception was *A. geminorum*, a species recently separated from *A. pittii* [7]. Fourteen isolates formed a monophyletic clade and had complete nucleotide sequence identity with the *A. geminorum* type strain J00019, which, however, was poorly discriminated from *A. pittii* (nucleotide distance 0.003; bootstrap value 67%). The two species *A. oleivorans* (https://lpsn.dsmz.de/species/acinetobacter-oleivorans, accessed on 16 July 2023) and *A. septicus* (https://lpsn.dsmz.de/species/acinetobacter-septicus, accessed on 16 July 2023), which have the nomenclatural status “not validly published”, were considered the same species, respectively, as *A. calcoaceticus* (*A. calcoaceticus/oleivorans*) and *A. ursingii* (*A. ursingii/septicus*). The latter two species/groups, as well as *A. johnsonii* and *A. lwoffii*, showed the highest intraspecies variability in *rpoB* gene sequences, with a maximum nucleotide divergence between strains of the same species of 0.049, 0.029, 0.027, and 0.033, respectively (Appendix A: Pairwise nucleotide distance matrix of partial *rpoB* sequences of the studied clinical isolates and reference strains).

*A. pittii* was the major *Anb* species, with 100 isolates that were geographically scattered. Three other species of the *Acb* complex (*A. nosocomialis*, *A. calcoaceticus/oleivorans*, and *A. geminorum*) and one species (*A. bereziniae*) that does not belong to the *Acb* complex each included more than 10 isolates from diverse geographic sites. The nine remaining *Anb* species (*A. junii*, *A. soli*, *A. seifertii*, *A. johnsonii*, *A. lwoffii*, *A. ursingii/septicus*, *A. haemolyticus*, *A. radioresistens*, and *A. courvalinii*) comprised one to nine isolates each. Curiously, one isolate, M19-2435, showed the highest nucleotide sequence identity (>99.4%) with the reference strains of *A. courvalinii*, a soil-dwelling species that had not previously been isolated from humans [13,44,45]. This isolate was recovered from a urine sample of an eight-year-old male patient with a neurogenic bladder after he was operated at the Children’s Hospital of Petrozavodsk, Russia, in April 2019.

### 2.2. Accuracy of Anb Species Identification Using MALDI-TOF MS Systems

Table 1 summarizes the results of the *Anb* species identification using two MALDI-TOF MS systems, the Autobio Autof and the Bruker Biotyper, compared to the reference method of *rpoB* gene sequencing. Figure 1 details the results for the individual isolates. Since *A. oleivorans* [9] has the nomenclatural status “not validly published”, it was considered the same species as *A. calcoaceticus* (*A. calcoaceticus/oleivorans*), and the identification of *A. oleivorans* as *A. calcoaceticus* was not considered incorrect. The overall rates of correct species identification were 189/234 (80.8%) for Autobio and 207/234 (88.5%) for Bruker. Both systems provided confident species-level identification of all isolates of *A. haemolyticus*, *A. johnsonii*, *A. junii*, *A. lwoffii*, *A. radioresistens*, *A. soli*, and *A. ursingii*. In addition, Autobio correctly identified all isolates of *A. nosocomialis* and *A. pittii*, two and eleven of which were misidentified by Bruker, respectively, as *A. baumannii* (log scores: 2.15–2.3) and *A. lactucae* (log scores: 1.99–2.26). On the other hand, Bruker correctly identified all isolates of *A. bereziniae*, *A. calcoaceticus/oleivorans* (as *A. calcoaceticus*), *A. courvalinii*, and *A. seifertii*, while Autobio misidentified all isolates of *A. bereziniae* as *A. guillouiae* (scores: 9.416–9.705), *A. seifertii* as *A. pittii* (scores: 8.990–9.348), a single isolate of *A. courvalinii* as *A. haemolyticus* (score 9.181); additionally, it provided no confident identification for four isolates of *A. calcoaceticus/oleivorans* (scores: 4.437–5.688). Finally, both the Autobio and Bruker systems misidentified all isolates of *A. geminorum*, respectively, as *A. pittii* (scores: 7.445–9.607) and *A. lactucae* (log scores: 2.1–2.23). In order to improve the accuracy of the identification, we generated reference mass spectra of representative *Anb* isolates and added them to the user libraries. Subsequently, all *Anb* isolates from our collection were correctly identified to the species level when tested against both the proprietary and our custom spectra libraries (data not shown).

### 2.3. Infections Caused by Anb Species

The *Anb* isolates were most often isolated from clinical samples of hospitalized patients with lower respiratory tract infections (75/234 (32.1%)) and urinary tract infections (UTIs) (65/234 (27.8%)). Other primary sites of infection included, in order of decreasing abundance: heart and blood vessels, 43/234 (18.4%), the abdominal cavity, 25/234 (10.7%), skin and soft tissues, 18/234 (7.7%), bones and joints, 4/234 (1.7%), the central nervous system, 3/284 (1.3%), and eye appendages, 1/234 (0.4%). Overall, we did not find any association between particular *Anb* species and the site of infection. All species represented by more than one isolate were isolated from diverse infections (Figure 1). Only two species were statistically significantly more abundant in specific loci: 14/16 (87.5%) *A. oleivorans* isolates were recovered from UTIs (the proportion of UTI isolates among other species was 51/218 (23.4%), *p* = 0.0001, Fisher’s exact test with the Holm correction), and 7/9 (77.8%) *A. soli* isolates were recovered from primary bloodstream infections (the proportion of bloodstream isolates among other species was 36/225 (16.0%), *p* = 0.0001). It should be noted, however, that all UTI isolates of *A. oleivorans* were isolated in the same pediatric hospital in Petrozavodsk city between October 2016 and February 2019. Likewise, all bloodstream isolates of *A. soli* were isolated in the same neonatal care unit in Kazan city between January 2019 and July 2019. Therefore, a common source of infection could not be excluded in the case of both *A. oleivorans* UTI and *A. soli* bloodstream isolates.

In addition to these two examples of potentially clonal expansions of *A. oleivorans* and *A. soli*, we observed a few more “in-hospital clusters” of isolates of the same species. These included five clusters of 7 to 15 isolates of *A. pittii* in different hospitals and one cluster of 11 isolates of *A. nosocomialis*. Records indicated that 171/234 (73.1%) infections due to *Anb* species were considered nosocomial as they occurred more than 48 h after patient admission to the hospital or were thought to be acquired during previous hospital admissions.

### 2.4. Susceptibility to Antibiotics and the Presence of Acquired Carbapenemase Genes

Table 2 provides a statistical summary of the susceptibility of *Anb* isolates to 11 antibiotics, and Figure 1 additionally shows the resistance profiles of the individual isolates to key agents representing major antibiotic classes. Most *Anb* isolates were susceptible to all antibiotics tested. The resistance rates to all β-lactam antibiotics and sulbactam were below 6%, and resistance to β-lactams was not confined to any particular species. Importantly, resistance to carbapenems (imipenem or meropenem) was detected in 11 sporadic isolates: 5 *A. pittii*, 2 *A. bereziniae*, and 1 isolate each of *A. johnsonii*, *A. nosocomialis*, *A. oleivorans*, and *A. ursingii*, all of which, except *A. johnsonii*, were additionally resistant to antibiotics of at least two other classes. Thirteen (5.6%) isolates, including five *A. pittii*, four *A. bereziniae*, and one isolate each of *A. nosocomialis*, *A. oleivorans*, *A. seifertii*, and *A. ursingii*, were resistant to sulbactam at an MIC breakpoint of ≥16 mg/L. The overall resistance rates to non-β-lactam agents were: 8.7% to tobramycin, 9.9% to amikacin, 11.2% to gentamicin, 7.4% to ciprofloxacin, 8.7% to trimethoprim–sulfamethoxazole, and 3.0% to colistin. The seven colistin-resistant isolates belonged to three species: four *A. bereziniae*, two *A. pittii*, and one *A. seifertii*. Notably, *A. bereziniae* displayed significantly higher resistance rates than other *Anb* species to colistin (22.2% vs. 1.4%, *p* = 0.0008, Fisher’s exact test with the Holm correction), sulbactam (22.2% vs. 4.21%, *p* = 0.0118), aminoglycosides (66.7% vs. 7.9%, *p* = 0.0001), and trimethoprim–sulfamethoxazole (66.7% vs. 3.8%, *p* = 0.0001); it also showed higher resistance levels to tigecycline (MIC_90%_: ≥16 mg/L vs. 1 mg/L).

Acquired carbapenemase genes were detected in 4/234 (1.7%) isolates collected at geographically distant sites. One isolate each of *A. bereziniae* and *A. pittii* carried the *bla*_NDM_ gene, another isolate of *A. pittii* carried the *bla*_OXA-58-like_ gene, and one isolate of *A. johnsonii* harbored the *bla*_NDM_ and *bla*_OXA-58-like_ genes simultaneously. The carbapenemase-positive isolates of *A. bereziniae* and *A. pittii* displayed high-level resistance to imipenem and meropenem, with MICs of ≥16 mg/L. Curiously, however, the double-carbapenemase-positive isolate of *A. johnsonii* showed borderline resistance to imipenem, with an MIC of 8 mg/L, and susceptibility to meropenem with an MIC of 1 mg/L. Finally, as expected, the isolates of *A. radioresistens*, which were susceptible to carbapenems, tested positive by PCR for the *bla*_OXA-23-like_ gene, which is known to be intrinsic chromosomally-encoded and usually weakly expressed in this species [46].

## 3. Discussion

This study characterized 234 clinical *Anb* isolates collected from hospitalized patients in multiple centers across Russia and Kazakhstan over the last seven years. These isolates belonged to 14 different species, as was determined by *rpoB* gene sequencing, and jointly comprised 6.2% of *Acinetobacter* spp. (the remaining isolates were *A. baumannii*) and 0.7% of all bacterial isolates collected as part of the sentinel surveillance program [47]. Among the *Anb* species, the most abundant was *A. pittii* (42.7%), followed by *A. nosocomialis* (13.7%), *A. calcoaceticus/oleivorans* (9.0%), *A. bereziniae* (7.7%), and *A. geminorum* (6.0%). *A. pittii* and *A. nosocomialis* are both known to be the next most common *Acinetobacter* spp. after *A. baumannii* to cause nosocomial infections [20]. *A. geminorum*, which has recently been separated from *A. pittii* [7], has a highly similar *rpoB* gene sequence. On the contrary, *A. oleivorans*, which has not been assigned the nomenclatural status of a validly published named species, and which was therefore regarded in our analysis as one species with *A. calcoaceticus*, has a more divergent *rpoB* gene sequence. It is worth mentioning that the *rpoB* sequences of 16 isolates of the *A. calcoaceticus/oleivorans* group clustered closely with those of the reference strains of *A. oleivorans*, while only 5 isolates clustered with the reference strains of *A. calcoaceticus.* This is an unexpected finding given that *A. oleivorans* has commonly been described as a soil-dwelling carbohydrate-hydrolyzing bacterium and has not been associated with humans [9,48,49,50]. Another interesting finding was the isolation of *A. courvalinii* [44,45] from a urine clinical sample of an eight-year-old male patient with cystitis, which, to our knowledge, represents the first documented case of human infection with this species.

The studied *Anb* isolates were recovered from different primary sites of infection, mostly the lower respiratory tract, urinary system, and blood stream, with no particular association between the infection locus and the species. It is worth noting that *A. oleivorans* and *A. soli* were isolated significantly more often, respectively, from urine and blood. However, all the urine isolates of *A. oleivorans* were collected in the same hospital and could therefore have a common origin, as were all the blood isolates of *A. soli*. In addition to these two examples, we observed several in-hospital clusters of infections due to *A. pittii* and *A. nosocomialis*, indicating their likely nosocomial transmission.

One of the most important tasks of this study was to assess the performance of two MALDI-TOF MS systems for the identification of *Anb* species. The Autobio and Bruker systems demonstrated the overall rates of correct species-level identification, respectively, of 80.8% and 88.5% for our collection of *Anb* isolates. These rates are lower than those reported from the study of the routine collection of bacterial isolates of any species (>97%) [51]. However, our results correspond well with the findings of many studies that focus specifically on the MALDI-TOF MS identification of *Acinetobacter* spp. [21,26,27]. In line with these studies, we found that most cases of misidentification were likely due to the absence or insufficient number of mass spectra of new species in the reference libraries, as was the case for *A. geminorum* in the Autobio and Bruker libraries and *A. courvalinii* in the Autobio library; the misidentifications could also be due to the potentially inconsistent species identification of the reference strains present in the library, as was the case for *A. bereziniae* and *A. seifertii* being identified with high scores as *A. guillouiae* and *A. pittii*, respectively, by Autobio. The low-score identification results of single isolates of *A. calcoaceticus/oleivorans*, *A. nosocomialis*, and *A. pittii* were also likely due to the small number of mass spectra of these species in the reference libraries. Importantly, however, we found that all identification errors could be eliminated by adding the spectra of “difficult” strains, correctly identified using the reference method, to the custom user libraries. These findings once again highlight the importance of maintaining the reference spectra libraries in accordance with current taxonomy and nomenclature of *Acinetobacter* spp. and the need to include strains of newly described species in the libraries.

The studied *Anb* isolates were mostly susceptible to antibiotics. The resistance rates to all agents did not exceed 10% (except for gentamicin, at 11.2%). These rates are significantly lower than those reported for *A. baumannii* isolates from the same surveillance program [47,52,53]. For example, the prevalence of resistance to trimethoprim–sulfamethoxazole, aminoglycosides, ciprofloxacin, and carbapenems was 6 to 20 times lower in *Anb* than in *A. baumannii* (https://amrmap.net/?id=BWSMM21DC29DC11, accessed on 16 July 2023) [47]. The exception was resistance to colistin, which was more than three times higher in *Anb* than in *A. baumannii* (3.0% vs. 0.8%), with most colistin-resistant isolates being identified as *A. bereziniae*. The latter species was also more resistant to sulbactam, aminoglycosides, trimethoprim–sulfamethoxazole, and tigecycline than other *Anb* species. The acquired resistance to colistin in *A. bereziniae* has previously been found to be associated with mutations in the *pmrB* gene involved in lipopolysaccharide modification [54]. Resistance to most aminoglycosides, including amikacin, has been attributed to plasmid-carried genes for aminoglycoside-modifying enzymes (*aphA6*, *aac(6′)-31* and *aadA1*) [36,55,56].

The primary mechanisms of resistance to carbapenems in *Acinetobacter* spp. include the overexpression of naturally occurring species-specific carbapenemases of molecular class D (OXA) [57] and the production of acquired carbapenemases of class D (mainly OXA-23-, OXA-24/40-, and OXA-58-like enzymes) and class B (mainly NDM and IMP) [4]. Our study revealed the low prevalence of acquired carbapenemases in *Anb* isolates (1.7%) as compared to *A. baumannii* (81.9%) (https://amrmap.net/?id=AN0Fl54mX14mX14, accessed on 16 July 2023). Moreover, only OXA-58-like and NDM carbapenemases were found alone or in combination in sporadic isolates of *A. bereziniae*, *A. johnsonii*, and *A. pittii*, whereas, in *A. baumannii* in Russia, these carbapenemases were rare, and the OXA-23- and OXA-24/40-types were the predominant carbapenemases detected [47,53]. Likewise, studies from many other countries have found the same differences in the frequency of various acquired carbapenemases in *A. baumannii* and *Anb*. OXA-58 has been detected in the early carbapenem-resistant isolates of *A. baumannii* in Europe, Asia, and South America [58,59] and, recently, mostly in *Anb* species, including *A. bereziniae*, *A. colistiniresistens*, *A. johnsonii*, *A. junii*, *A. nosocomialis*, *A. pittii*, *A. radioresistens*, *A. seifertii*, *A. towneri*, and *A. ursingii*, from all over the world [37,60,61,62,63,64,65,66,67,68,69]. Similarly, metallo-β-lactamases, mostly NDM-1 and IMP variants, have been found in global *Anb* isolates, often in combination with OXA-58 [62,63,64,70,71,72,73,74,75,76,77,78].

While our study covered a large geographic area and a long time period, it did not identify and include some of the *Anb* species that have been previously cultured from humans. This is an obvious limitation of the study, which demonstrates the importance of ongoing surveillance and research. Our further study will utilize whole-genome sequencing data to infer the phylogenetic relationships between closely related *Acinetobacter* spp., such as *A. pittii* and *A. geminorum*, and to explore in greater depth the mechanisms of antibiotic resistance in *Anb* isolates. Further studies will also be needed to continuously evaluate the performance of MALDI-TOF MS platforms with updated spectra libraries for identification of *Acinetobacter* spp.

## 4. Materials and Methods

### 4.1. Bacterial Isolates

All the studied isolates and accompanying metadata were collected as part of the national sentinel surveillance program of antimicrobial resistance (AMR) in bacterial pathogens isolated from hospitalized patients [47]. Between 1 January 2016 and 31 December 2022, 94 participating hospitals from 42 cities across Russia and 1 hospital in Karaganda, Kazakhstan, contributed a total of 34,524 non-duplicate (one per patient/case of infection) clinical bacterial isolates, of which 3754 (10.9%) were confirmed in the central surveillance laboratory as *Acinetobacter* species. The isolates were recovered from representative clinical specimens (blood, tissue biopsies, cerebrospinal fluid, bronchoalveolar lavage, sputum, urine, etc.) of patients with clinical symptoms of infections; isolates from surface swabs, screenings, and environmental samples were excluded. All isolates with accompanying clinical and epidemiological information were referred to the central surveillance laboratory of the Institute of Antimicrobial Chemotherapy (IAC), where species identification, antibiotic susceptibility testing, and molecular genetic characterizations of the isolates were performed.

### 4.2. Species Identification with MALDI-TOF MS

The bacterial isolates were identified with MALDI-TOF MS, using both the Microflex LT-MALDI Biotyper System with the Biotyper spectral database ver.11 (Bruker Daltonics, Bremen, Germany) and the Autof MS2000 System with the current online spectral library (Autobio Diagnostics, Zhengzhou, China). The standard methods of direct colony transfer and rapid (on-target) extraction with formic acid were used for sample preparation prior to the acquisition of mass spectra, as described previously [79,80]. The criteria for “confident species-level identification” were as follows: log scores ≥ 2.0 for the Bruker Biotyper System and scores ≥ 9.0 for the Autobio Autof System. Only the highest-match score values (1st of the 10 best hits) of the clinical isolates against the corresponding reference libraries were reported in the Results section. To generate the reference spectra of the selected *Acinetobacter* strains for the Autobio library, the full (in-tube) extraction method was used; the extracts from each culture were spotted four times onto a ground steel target and each spot was measured and processed three times according to the manufacturer’s instruction.

### 4.3. Additional Tests to Distinguish A. baumannii

*A. baumannii* isolates were additionally distinguished by the detection of species-specific *bla*_OXA-51-like_ genes using real-time PCR [24] and by multilocus sequencing typing (MLST) using the University of Oxford and the Institute Pasteur typing schemes [81,82].

### 4.4. Sequencing of the rpoB Gene and Analysis of the Sequence Data

All isolates preliminarily identified as *Anb* species were subjected to *rpoB* gene sequencing, which was used a reference identification method [21]. Briefly, a 397 bp internal fragment of the *rpoB* gene was amplified by PCR using the primers Ac696F, 5′-TAYCGYAAAGAYTTGAAAGAAG-3′, and Ac1093R, 5′-CMACACCYTTGTTMCCRTGA-3′, as described elsewhere [83]. The PCR products were purified with exonuclease I and a shrimp alkaline phosphatase treatment and were sequenced on both strands using the same primers and a BigDye Terminator v3.1 Cycle Sequencing Kit (Thermo Fisher Scientific, Waltham, MA, USA). The sequencing products were analyzed using a Applied Biosystems 3500 Genetic Analyzer (Life Technologies, Carlsbad, CA, USA). Quality assessments and the assembly of sequences were performed using a QIAGEN CLC Genomics Workbench 21.0 (QIAGEN, Aarhus, Denmark).

The *rpoB* gene sequences of the reference and type strains (one to three per species) of 27 *Acinetobacter* spp. (*A. baumannii*, *A. baylyi*, *A. beijerinckii*, *A. bereziniae*, *A. calcoaceticus*, *A. colistiniresistens*, *A. courvalinii*, *A. geminorum*, *A. guillouiae*, *A. haemolyticus*, *A. johnsonii*, *A. junii*, *A. lactucae*, *A. lwoffii*, *A. nosocomialis*, *A. oleivorans*, *A. parvus*, *A. pittii*, *A. radioresistens*, *A. schindleri*, *A. seifertii*, *A. septicus*, *A. soli*, *A. towneri*, *A. ursingii*, *A. variabilis*, and *A. venetianus*) were obtained from the NCBI GenBank (https://www.ncbi.nlm.nih.gov/genbank/, accessed on 16 July 2023) and the ATCC Genome Portal (https://genomes.atcc.org/, accessed on 16 July 2023). The *rpoB* gene sequences of the studied clinical isolates and reference strains were aligned together (after trimming the primer sites) and the resulting alignment was used to produce a maximum-likelihood tree with the Tamura–Nei model and 1000 bootstraps using MEGA v.11.0.13 [84]. The visualization and annotation of the phylogenetic tree were performed using iTOL v.6.7.6 [85].

### 4.5. Antimicrobial Susceptibility Testing

Antimicrobial susceptibility testing (AST) was conducted for the following agents: amikacin, cefepime, ciprofloxacin, colistin, gentamicin, imipenem, meropenem, sulbactam, tigecycline, tobramycin, and trimethoprim–sulfamethoxazole (1:19). This was accomplished by using a reference broth microdilution method according to ISO 20776-2:2021 [86] and the methodology of the European committee on antimicrobial susceptibility testing (EUCAST) [87]. *Escherichia coli* ATCC 25922, *E. coli* NCTC 13846, and *Pseudomonas aeruginosa* ATCC 27853 were used as quality control strains for AST. The AST results were interpreted according to the EUCAST v.13.0 Clinical Breakpoints [88] for all agents except for cefepime and sulbactam, for which the results were interpreted using the CLSI M100 ED33 breakpoints [89], and for tigecycline, for which only the MIC_50_ and MIC_90_ values were reported due to lack of EUCAST and CLSI interpretive criteria.

### 4.6. Real-Time PCR Detection of Carbapenemase Genes

The detection of genes encoding the most common *Acinetobacter*-spp.-acquired carbapenemases of molecular class D (OXA-23-, OXA-24/40-, and OXA-58-like) and class B (metallo-β-lactamases: NDM, IMP and VIM) was performed using commercial real-time PCR assays: AmpliSens MDR Acinetobacter-OXA-FL and AmpliSens MDR MBL-FL (Central research institute of epidemiology, Moscow, Russia), according to manufacturer’s instructions. Amplification reactions were carried out using the DTPrime 5X1 Real-Time PCR System (DNA Technology, Moscow, Russia). Strains of *A. baumannii*, *A. pittii*, and *P. aeruginosa* carrying the genes of the known carbapenemases from the IAC collection [47] were used as positive controls.

### 4.7. Data Availability

The DNA sequences obtained in this study were deposited in the European Nucleotide Archive under the project accession number PRJEB61953. The clinical, epidemiological, geospatial, and temporal characteristics of the studied *Anb* isolates, the quantitative (MIC) and qualitative (categorical) AST data, the carbepenemase gene status, and the *rpoB* phylogenetic clustering data were made available as an open-access project at Microreact [90]: https://microreact.org/project/fKrejDn6vqcervyWwMAs8W-acinetobacter-non-baumannii-species (accessed on 16 July 2023).

## 5. Conclusions

This study represents one of the largest surveys of *Anb* isolates collected from infections in hospitalized patients. It provides new insights into the methods of identification, as well as the occurrence, species distribution, and antibiotic resistance traits of *Anb* isolates. It also reports for the first time the isolation of *A. oleivorans* and *A. courvalinii* from clinical samples of patients with urinary tract infections.

## Figures and Tables

**Figure 1 antibiotics-12-01301-f001:**
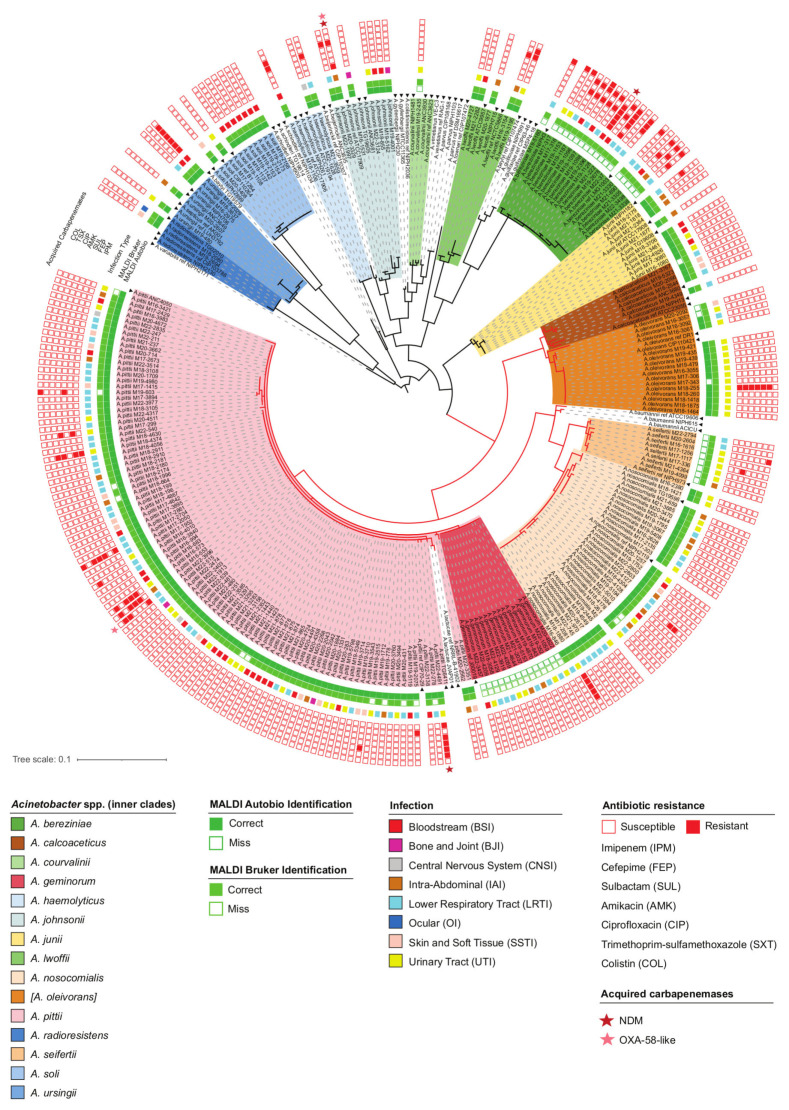
Midpoint-rooted maximum-likelihood distance tree of partial *rpoB* sequences from clinical isolates and reference strains of *Anb* species. The various species identified in this study are shown with the colored inner clades, according to the legend. Reference strains are marked with black triangles. Species of the *A. calcoaceticus-baumannii* complex are shown with red branches. Concentric rings show (from the center to the outside): (i) the accuracy of the species identification with the Autobio and Bruker MALDI-TOF MS systems; (ii) the infection type; (iii) the antibiotic resistance profiles; and (iv) the presence of acquired carbapenemase genes (see legends).

**Table 1 antibiotics-12-01301-t001:** Accuracy of the species identification of *Anb* isolates using MALDI-TOF MS systems.

*Anb* Species ^1^	No. of Isolates	No. (Percent) Correctly Identified	
		Autobio Autof	Bruker Biotyper
*A. bereziniae*	18	0 (0%)	18 (100%)
*A. calcoaceticus/oleivorans*	21	17 (81%)	21 (100%)
*A. courvalinii*	1	0 (0%)	1 (100%)
*A. geminorum*	14	0 (0%)	0 (0%)
*A. haemolyticus*	3	3 (100%)	3 (100%)
*A. johnsonii*	7	7 (100%)	7 (100%)
*A. junii*	9	9 (100%)	9 (100%)
*A. lwoffii*	5	5 (100%)	5 (100%)
*A. nosocomialis*	32	32 (100%)	30 (93.8%)
*A. pittii*	100	100 (100%)	89 (89%)
*A. radioresistens*	2	2 (100%)	2 (100%)
*A. seifertii*	8	0 (0%)	8 (100%)
*A. soli*	9	9 (100%)	9 (100%)
*A. ursingii/septicus*	5	5 (100%)	5 (100%)

^1^ as determined by *rpoB* gene sequence analysis.

**Table 2 antibiotics-12-01301-t002:** Minimum inhibitory concentration (MIC) range, MIC50, MIC90, and proportion of *Anb* isolates by category of susceptibility to 11 antibiotics.

Antibiotic	MIC, mg/L			Percent by Category ^1^		
	Range	50%	90%	S	I	R
Amikacin	≤0.5–≥256	1	8	90.1	-	9.9
Cefepime ^2^	≤0.5–≥256	2	16	88.7	5.7	5.7
Ciprofloxacin	≤0.06–≥128	0.125	0.5	-	92.6	7.4
Colistin	≤0.06–≥64	0.5	2	97.0	-	3.0
Gentamicin	≤0.25–≥256	0.5	8	88.8	-	11.2
Imipenem	≤0.06–≥128	0.125	0.5	95.7	-	4.3
Meropenem	≤0.06–≥128	0.25	0.5	95.2	0.9	3.9
Sulbactam ^3^	≤0.25–≥256	1	4	92.2	2.2	5.6
Tigecycline ^4^	≤0.03–≥16	0.25	1	-	-	-
Tobramycin	≤0.25–≥256	0.25	2	91.3	-	8.7
Trimethoprim– sulfamethoxazole (1:19) ^5^	≤0.125–≥256	0.125	4	89.6	1.7	8.7

^1^ S, susceptible; I, susceptible, increased exposure (EUCAST), or intermediate resistant (CLSI); R, resistant. EUCAST v13.0 MIC clinical breakpoints were used unless otherwise stated. ^2^ For cefepime, CLSI M100 ED33 MIC breakpoints were used. ^3^ For sulbactam, the MIC breakpoints (S ≤ 4 mg/L, I 8 mg/L, R ≥ 16 mg/L) were based on CLSI M100 ED33 ampicillin–sulbactam (2:1) MIC breakpoints (S ≤ 8/4 mg/L, I 16/8 mg/L, R ≥ 32/16 mg/L), where sulbactam comprises the active component of the combination against Acinetobacter. ^4^ For tigecycline, no MIC breakpoints were established by either EUCAST or CLSI. ^5^ MIC values refer to trimethoprim.

## Data Availability

The DNA sequences obtained in this study were deposited in the European Nucleotide Archive under the project accession number PRJEB61953. The clinical, epidemiological, geospatial, and temporal characteristics of studied Anb isolates, the quantitative (MIC) and qualitative (categorical) AST data, the carbepenemase gene status, and the *rpoB* phylogenetic clustering data were made available as an open-access project at Microreact (https://microreact.org/project/fKrejDn6vqcervyWwMAs8W-acinetobacter-non-baumannii-species, accessed on 16 July 2023).

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
