# Peer review of "Acinetobacter Non-baumannii Species: Occurrence in Infections in Hospitalized Patients, Identification, and Antibiotic Resistance"

_antibiotics, 2023, doi:10.3390/antibiotics12081301_

Round 1

Reviewer 1 Report

This manuscript describes the analysis of over 3700 isolates of Anb Acinetobacter isolates from Kazakhstan and Russia. The isolates were moslty clinical isolates. Analyses included speciation by MALDI-Tof MS with two systems, speciation by rpoB sequencing, susceptibility testing, and carbapenemase gene detection by RT-PCR. The results demonstrated that MALDI-Tof-MS identification is only possible with databases containing the appropriate species and with enough spectra from those species. They also identified a species that is significantly more MDR that the other Anb species studied. Overal, the study was well done, the paper well written, and the conclusions were supported by the data. This paper can serve as a reference material for the Anb isolates from this region of the world and for the infections they cause. Some minor comments follow that the authors can consider addressing:

1. Line 61, define ESKAPE

2. Line 261, I believe you mean noting rather than nothing, I suggest not using autocorrect when writing a manuscript. 

Author Response

Dear Reviewer,

We are very grateful for your review and positive evaluation of our manuscript. We updated the manuscript in accordance with your suggestions. In the revised version, the changes are highlighted in yellow. The following is a point-by-point response to the comments:

  1. Line 61, define ESKAPE

We defined the ESKAPE.

  1. Line 261, I believe you mean noting rather than nothing, I suggest not using autocorrect when writing a manuscript.

Thank you very much. We corrected the sentence.

Reviewer 2 Report

The manuscript was well written. The abstract, introduction, result and discussion are good. 

I would like to suggest certain points.

1. Not only rpo B gene used for differentiation between the Acenitobacter spp., there have been other genes such as oxa, gyraseB, recA etc. Why did the authors choose the rpoB?

2. It would be nice to have the supplement reporting the nucleotide similarity between the reference strains and the tested strains. 

3. The authors reveal phylogenetic tree based on the rpoB gene. It would be nice to discuss more on the tree. (the author discuss some species but not al).

4. What is the limitation of the study.

5. The conclusion answered all objectives. However, the authors should add plans for further study based on the discovery.

Author Response

Dear Reviewer,

We are very grateful for your review and valuable comments. We updated the manuscript in accordance with your suggestions. In the revised version, the changes are highlighted in yellow. The following is a point-by-point response to your comments:

  1. Not only rpo B gene used for differentiation between the Acenitobacter spp., there have been other genes such as oxa, gyraseB, recA etc. Why did the authors choose the rpoB?

We agree with your comment. However, in the Introduction section, we already acknowledged the fact that sequencing of several other genes, including gyrB, recA and blaOXAs, has been used for differentiation of Acinetobacter spp. We also emphasized that sequencing of rpoB has been most often used for this purpose. We selected rpoB sequencing because it has been extensively evaluated with any Acinetobacter spp., both established and newly described, and has been suggested as a reference method in the cited literature [8, 21-23, 83]. Amplification of species-specific blaOXA genes requires the use of multiple primer pairs, besides these loci has been studied only with selected, not all Acinetobacter spp. Other loci (e.g. 16S rRNA gene) have been shown to provide insufficient discrimination within Acb complex, or have been found to be more affected by HGT and intragenic recombination (e.g. gyrB).

  1. It would be nice to have the supplement reporting the nucleotide similarity between the reference strains and the tested strains.

Thank you for this valuable suggestion. We have added a Supplementary Table S1: Pairwise nucleotide distance matrix of partial rpoB sequences of studied clinical isolates and reference strains.

  1. The authors reveal phylogenetic tree based on the rpoB gene. It would be nice to discuss more on the tree. (the author discuss some species but not al).

We presented phylogenetic tree in Figure 1 with the primary aim to show the clustering of rpoB gene sequences of the clinical isolates and reference strains. We tried to keep the paper concise and easy to read and avoided extensive description of Acinetobacter phylogeny based on rpoB gene because it has been already widely reported. Following your recommendation, however, we added the sentences (highlighted in yellow) in the first paragraph of the Results section about the maximum intraspecies variability of rpoB gene sequences and added the reference to Supplementary Table S1.

  1. What is the limitation of the study.

See below Q5.

  1. The conclusion answered all objectives. However, the authors should add plans for further study based on the discovery.

Our surveillance was limited to certain geographic area and time and did not identify some human-associated Anb species described earlier, which is an obvious limitation. We added the following paragraph (highlighted in yellow) at the end of the Discussion section:

"While our study spanned a large geographic area and a long time period, it did not identify and include some of the Anb species that have been previously cultured from humans. This is an obvious limitation of the study that stress the importance of ongoing surveillance and research. Our further study will utilize whole-genome sequencing data to infer phylogenetic relationship between closely related Acinetobacter spp., such as A. pittii and A. geminorum, and to explore in depth the mechanisms of antibiotic resistance in Anb isolates. Further studies will be also needed to continuously evaluate the performance of Acinetobacter spp. identification using MALDI-TOF MS systems with updated spectra libraries."

Thank you once again for reviewing our manuscript.